# Water Oriented City—A '5 Scales' System of Blue and Green Infrastructure in Sponge Cities Supporting the Retention of the Urban Fabric

Anna Zaręba [1,*] , Alicja Krzemińska [1] , Mariusz Adynkiewicz-Piragas [2] , Krzysztof Widawski [1], Dan van der Horst [3], Francisco Grijalva [4] and Rogelio Monreal [4]

1   Faculty of Earth Sciences and Environmental Management, University of Wrocław, ul. Cybulskiego 30, 50-205 Wrocław, Poland
2   Institute of Meteorology and Water Management National Research Institute (IMGW-PIB), ul. Podleśna 61, 01-673 Warszawa, Poland
3   School of Geosciences, University of Edinburgh, Drummond Street, Edinburgh EH8 9XP, UK
4   Department of Geology, University of Sonora, Campus Universitario Edif. 3C, Hermosillo 83000, Mexico
*   Correspondence: anna.zareba@uwr.edu.pl

**Abstract:** The article presented methods of urban development in terms of the application of the 'sponge city' concept, as well as the possibility of introducing different hydro-engineering solutions into the urban fabric that allow infiltration and retention at various scales of spatial planning. The aim of the paper was to indicate which specific solutions can be used in the city in multi-dimensional and multi-functional systems. As a result of the research, the concept of a '5-scales' diffusion of blue-green infrastructure elements was presented. Elements of this system are based on multi-scale blue-green infrastructure, creating a patchwork of 'blue connections' that fit into the city 'green' natural system and have a connection with urban rainwater drainage. These five elements together allow for the infiltration and retention of rainwater, and can be used in the design of ecologically sustainable water-oriented cities in the future.

**Keywords:** sponge city; water retention in the urban fabric; multi-scale design; blue-green infrastructure





## 1. Introduction

The importance of water retention in the urban fabric has long been as important as it is today. Intensive urban development, the sealing of a city's surface, irresponsible management of biologically active areas, as well as the indirect regulation of rivers, contribute to increasing surface runoff and have disturbed the water balance [1,2]. Climate change and the increasing incidence of extreme climate events have further increased the likelihood of flooding and drought [3–7]. In order to counteract these adverse phenomena, cities strive for new solutions using green areas and promoting innovative ideas for blue-green infrastructure. The spectrum of investment in this area is constantly growing, and emerging new technological innovations include revitalization of watercourses, the introduction of water retention reservoirs, as well as applications of smaller blue-green infrastructure elements such as rain gardens, tanks and bioswales, infiltration basins, etc. Such developments are aimed not only at supporting the slowing down of rainwater runoff by enabling natural infiltration, but also its retention and reuse. This at the same time significantly affects habitat quality and biodiversity and settlement units, which are characterized by the use of a closed cycle of hydrodynamics, and natural processes are created. Particularly important from the perspective of a city's needs is development based on the concept of a closed water cycle. It is assumed that cities in the near future will be able not only to create quasi-stable systems that will simulate and stimulate natural processes, but like so-called 'sponge cities' will 'deliver water' during droughts to where there is a need. Planning and appropriate selection of such solutions in the coming decades will be one of the challenges that many

cities will take on regardless of climate warming. These challenges can be addressed on a variety of scales, from the micro (street, square) to macro-scale solutions involving entire cities and their suburbs. The aim of the paper was to present a new '5-scales' diffusion model of blue-green infrastructure elements allowing for the retention and infiltration of water in the city at various scales of spatial development.

## 2. Materials and Methods

### 2.1. Methods

As part of the research, an analysis of the literature on rainwater retention and infiltration in the urban fabric was undertaken. Methods of spatial development of urban areas using blue-green infrastructure were examined, as well as the possibility of introducing hydro-engineering solutions into the urban fabric at various scales of spatial development from the micro (single-family house), through the meso (neighborhood: in a linear form at the scale of the street, and on the surface, at the scale of the residential district) to the macro scale for the entire city. Our intention was to indicate which specific solutions can be used at all scales in multi-dimensional and multi-functional systems. For the purposes of the article, the following analyses were carried out (Figure 1):

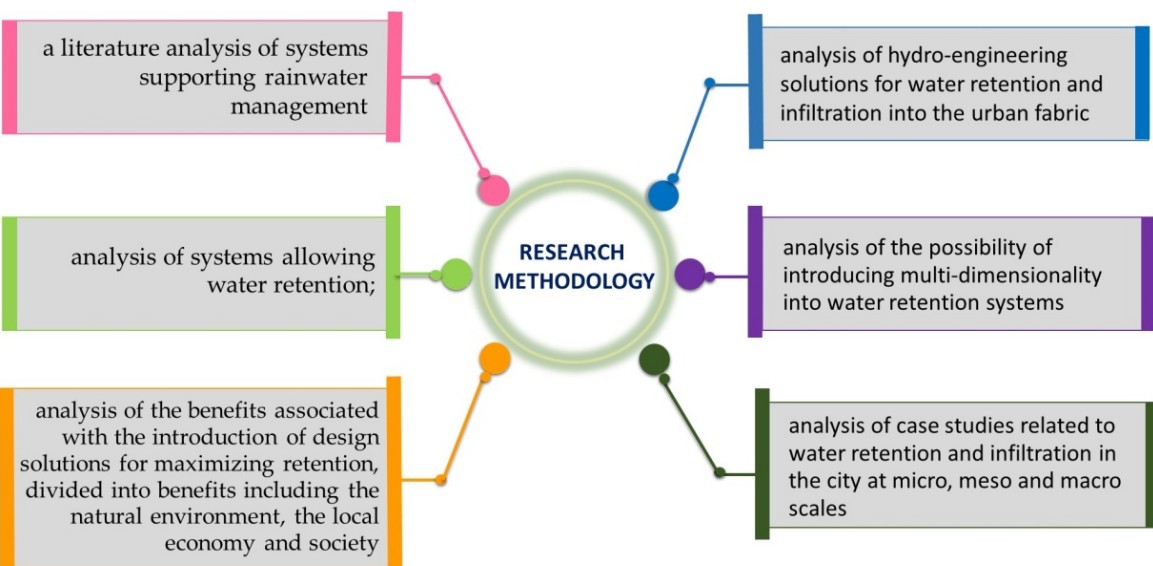

**Figure 1.** Research methodology concept diagram (own elaboration).

### 2.2. The Benefits of Supporting Sustainable and Ecological Rainwater Management: The Concept of 'Sponge Cities'

In the development of cities oriented towards natural water retention, the most important idea is to maintain a balance between permeable and impermeable areas. This sustainability contributes to the buffering of extreme phenomena, including flooding or drought, and is one of the basic activities helping the periodic retention of water, aiming to designate and protect the development of biologically active areas while supporting natural water infiltration. This protection should particularly apply to the river valleys that form the basis of the city's natural system. Of fundamental importance in terms of its functioning is to maintain ecological connections in blue-green infrastructure (including rivers, river valleys, and natural and artificial water reservoirs). Water retention in the city is also increased by the existence of diverse landforms, particularly natural depressions that can retain water, allow infiltration, and enable blue-green infrastructure investments, such as rain gardens, bioswales, infiltration basins, and many others. Only these macro-scale mutual relations are able to build a healthy biome for the new eco-city.

The 'Sponge City': A Concept Supporting Rainwater Management and Sustainable Development

The concept of the 'sponge city' is not new, but nowadays it should be included into the spatial planning system of an ecologically sustainable city. The design of 'sponge cities' is based on an innovative way of imitating and supporting the natural circulation of water in the urban environment, consisting of rainwater retention and purification thanks to the techniques used to install blue-green infrastructure which allow for rational water management in extreme conditions, i.e., drought and flooding. Under standard conditions, this system stabilizes the natural environment of the city and its biodiversity [8–18]. The innovative idea of the so-called 'sponge city' was presented for the first time at the 2012 Shenzhen International Forum on Low-Carbon Urban Development and Technology. This idea was so interesting that plans to build this type of city were formally announced at the Central Urbanization Working Conference and approved by Jinping Xi in 2013, and then introduced into city planning in China [10,19]. In 2015, the Ministry of Housing and Urban-Rural Development released the first comprehensive guide dedicated to the construction of sponge cities. By 2018, the next sponge city guide already contained more structured information, including but not limited to control of urban runoff, stormwater source control and implementation effectiveness, road surface flood control, urban water quality, ecological conservation and eco-system services, groundwater depth and condition, and urban heat island reduction [20,21]. It is also very important that the 'Guide' presents the first comprehensive definition of a sponge city, i.e., The Sponge City is a strategy for integrated urban water management. It is scientifically rooted in the laws of natural and social water cycles and their associated processes. It aims to mitigate urban waterlogging, control urban water pollution, and utilize rainwater resources as well as restore the ecological degradation of urban water' [12]. It was also pointed out that 'The Sponge City is based on large-scale infrastructure, including water pollution control, urban river restoration and waterlogging prevention; and on small-scale infrastructure such as sponge roads and sponge communities'. According to Wang [12], 'Sponge Cities are introduced to mitigate five urban water problems: water shortage, waterlogging, water pollution, ecological degradation and the 'city syndrome' (e.g., heat islands, turbidity islands and rain islands)' [12,22]. Zevenbergen et al. (2018) [7] listed three main benefits associated with the use of sponge cities: reducing economic losses (caused by floods), improving living conditions and creating an environment in which there would be investment opportunities related to the modernization of the hydro-engineering infrastructure and the introduction of new technologies related to water management (Figure 2).

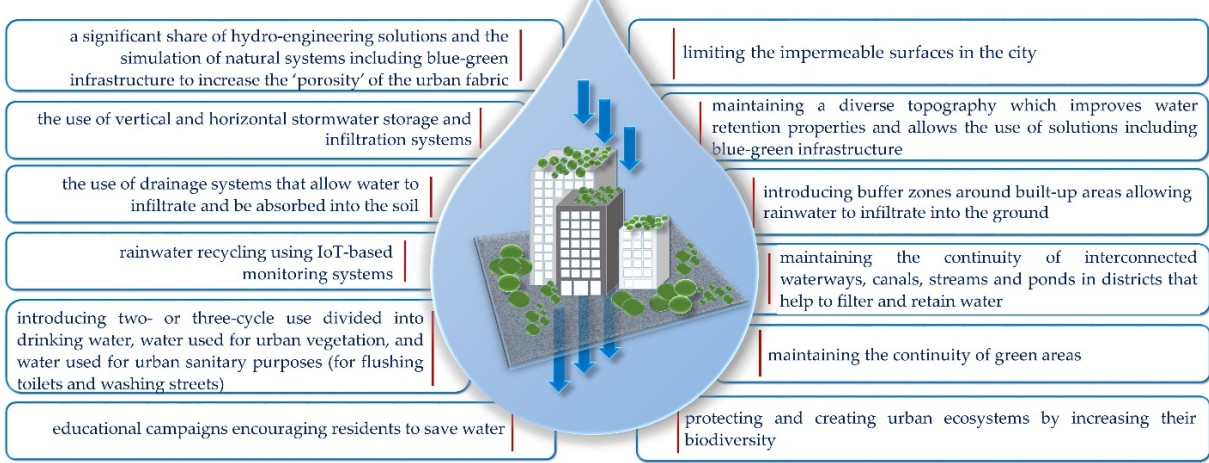

**Figure 2.** The principles on which the functioning of the sponge city is based (own elaboration).

Cities that rely on multi-variant solutions for blue-green infrastructure increase their retention possibilities, and thus their ecological potential. The benefits of blue-green

infrastructure implemented into the urban fabric are well described in the scientific world. In particular, a lot of research has been dedicated to the multi-dimensionality of blue-green infrastructure that affects environmental, social, and economic functions [2,23–27].

In contemporary urban design, there are many systems supporting appropriate and sustainable water management. These include, above all, sustainable urban drainage systems (SUDS), which include environmentally friendly technical solutions to drain rainwater from the city and reduce the negative impact of urbanization on surface water management [28–32]. The purpose of SUDS is to map the natural functions of the environment in relation to rainwater retention. One of the planning proposals helpful in shaping new housing estates is low impact development (LID). This is a concept whose origins date back to 1977 concerning the principles of design at the level of spatial planning taking into account the specificity of the landscape (e.g., landforms, geological structure, presence of aquatic and terrestrial ecosystems), which should form the basis for appropriate spatial management [33]. In this system, design is focused primarily on water management and the use of new areas including blue-green infrastructure to create an attractive landscape for housing estates [8,34,35]. The use of LID has been anchored in legislation in North America, including Canadaand the United States of America [22,36]. From the point of view of cities with increased water retention, among the more important concepts is water sensitive urban design (WSUD), which refers to the issue of a closed water cycle in the urban fabric using the imitation of natural processes in the water catchment. In the WSUD, the urban water cycle (mainly rainwater) is treated as similar to the natural one. The system also considers the use of other elements of the urban water cycle in the urban fabric. Rainwater, water supply and wastewater disposal systems, and aquatic ecosystems constitute here a framework for urban-scale design [22,37–41]. Other interesting concepts include sustainable urban water management (SUWM) initiated by the Swedish Foundation for Strategic Environmental Research (MISTRA) in 1999, which draws attention to five priorities that should guide modern cities: moving towards a non-toxic environment; improving health and hygiene; saving human resources; conserving natural resources; and saving financial resources [22,42]. An integrated urban drainage system (IUDS), a concept created in 1970 in Switzerland assumes the use of several systems related to water management in the city, including water treatment, distribution, sewerage and storm drainage, wastewater treatment, and environmental compartments [22,43]. All the water management systems in the city presented here are united by the so-called 'sponge city' concept, which uses a comprehensive design aimed at the protection and active use of rainwater in the city adapted from LID, SUDS, and WSUD, as well as the ancient Chinese concept of nature [44].

*2.3. Technologies Supporting the Construction of a Sustainable Blue-Green Infrastructure System in the Urban Fabric*

The existing technical solutions for blue-green infrastructure are mainly based on imitating nature. These are activities related primarily to planting vegetation [23,45,46] as well as the use of water retention in a filtration system of anthropogenic origin (e.g., filtration materials). Thanks to this combination, stable surface and underground filtration systems with adequate retention are formed. The choice of hydro-engeneering solution depends on specific environmental, physiographic, and spatial conditions. It can be both a system based on meadow vegetation, shrubs or tree stands, and/or the construction of filtration tanks of various sizes. An overview of the most important techniques to support filtration that are most commonly used is presented in Table 1. One of the most interesting solutions here are rain gardens increasingly often implemented in the landscape of modern cities. These combine the form of a garden and rainwater retention, which is then discharged into the ground (in the case of dry rain gardens located on permeable land). In the case of wet rain gardens (created on impermeable ground) and rain gardens in containers, their advantage is that they can be located directly next to buildings. The benefits of wet rain gardens include their possible use at various spatial scales from the

single-family plot, to a combined system of wet and dry rain gardens at the scale of the housing district as well as multiple micro applications across the entire city.

**Table 1.** Characteristics of natural and artificial filtration systems that can be used in the urban fabric (own elaboration).

| | Kind | Type | Description | Resources |
|---|---|---|---|---|
| **Natural Filtration Systems** | | | | |
| 1. | Riverside buffer vegetation | linear | Planted on the banks of water bodies, often through gabions connected to biogeochemical barriers. | [47–49] |
| 2. | Flower meadows on wetlands | surface | Multi-species flower meadows with compact root systems. Moisture-loving species supporting retention. | [50–52] |
| 3. | Parks and forests on wetlands | surface | Planted woody vegetation with different species composition suitable for wetlands. | [53,54] |
| **Ground Infiltration and Retention Systems** | | | | |
| 1. | Troughs and bioswales | linear | Absorption areas for linear drainage (most often located near roads)–the width of troughs should be from 1 to 2.5 m and their depth should be a minimum of 20 cm and a maximum of 20% of their width. | [15,46,55–60] |
| 2. | Infiltration basin | surface | Dry or wet in the form of channels closed with a damming structure (e.g., weir) covered with vegetation/for the accumulation of rainwater shallow depressions with a large area. | [57,59–64] |
| 3. | Infiltration tank/pond | surface | Tanks of different sizes, often planted with a mixture of grasses–the depth of the basin is from 0.3 to 1.0 m | [46,59,60,65–68] |
| 4. | Retention reservoirs | surface | Usually large earth-bound reservoirs, located in natural depressions and planted with vegetation whose root systems support the construction of the reservoir and consisting of plant species occurring locally (underwater, marsh and terrestrial vegetation) in order to retain water and then gradually discharge it into the sewage network, treatment plant or receiver: dry and wet retention, wet extended retention with additional capacity, micropool extended retention (with additional micropools), multiple reservoirs. | [45,46,59,66,67,69–72] |
| 5. | Stormwater wetlands/ constructed wetlands | surface | Wetland systems designed to treat contaminated rainwater through several mechanisms, including sedimentation, biological absorption, photodegradation and microbial decomposition, typically include shallow and deep areas to basins, meandering small watercourses and wetland vegetation to remove contaminants, shallow, extended retention and pond systems. | [66,72–78] |

<div align="center">

**Table 1.** *Cont.*

</div>

| | Kind | Type | Description | Resources |
|---|---|---|---|---|
| 6. | Rain gardens | surface | Located in natural or artificial depressions supplied with rainwater from roofs, roads and parking lots, created from several filtering layers which play a stabilizing role for the roots of plants. | [4,79–87] |
| 7. | Green roofs | surface | Due to location on compact urban buildings they have a significant impact on the load of buildings, some of the most technologically and artificially monitored solutions in blue-green infrastructure, consist of many layers (e.g., vegetation, filtering, separating membrane and thermal layers), green roofs are divided into extensive roofs (with a thickness of less than 15 cm) and intensive roofs (with a thickness of more than 15 cm). | [88–95] |
| 8. | Green walls | surface | Classified as 'green infrastructure' such as green roofs, in fact due to the need to introduce drainage and irrigation, as well as the possibility of rainwater retention, among the valuable elements of blue-green infrastructure. | [60,96–98] |
| 9. | Hydrophytic ponds | surface | Vegetation-covered systems with extended retention time, permanently and to varying degrees saturated with water, primarily used to purify rainwater. Larger systems, thanks to high capacity and throughput, create urban hydrophytic treatment. Plants effectively remove pollutants from precipitation wastewater and increase sedimentation. One variety of hydrophytic treatment is a sedimentation and biofiltration system consisting of three separate chambers: intensive sedimentation of suspended material, biochemical capture of dissolved impurities and a hydrophytic pond water vegetation zone. | [60,99–101] |
| 10. | Permeable surfaces with drainage | surface | Mineral, mineral and resin, permeable concrete or openwork surfaces, i.e., geogrids or concrete gratings filled with grass or gravel used in parking lots and on roads ensuring the infiltration of water without surface collection, solutions additionally supported by drainage through additional devices (e.g., perforated pipes, underground tanks filled with gravel or absorption wells). | [60,76,102–110] |
| 11. | Rainwater retention for households | point | For instance, butts for collecting rainwater from roof guttersZOUYANThe four main types of residential drainage system are surface, subsurface, slope with downspouts and gutter. | [60,107,111–113] |

**Table 1.** *Cont.*

| | Kind | Type | Description | Resources |
|---|---|---|---|---|
| **Underground Infiltration Systems with Retention** | | | | |
| 1. | Absorption wells | point | Most often concrete underground wells without a floor, filled with filtration material with high permeability (e.g., gravel, stone grit), occupying a small space but providing little filtration. | [46,68,107,114–117] |
| 2. | Filtration trenches | linear | Linear underground filtering systems, filled with filtration material with high permeability, wells and absorption trenches are located on soils with low permeability, improving the filtration properties of a specific place. | [60,107,114–118] |
| 3. | Drainage system | surface | Drainage system and drainage chambers additionally supported by a layer of filtration materials, gravel, and stones, used on land with high permeability, helps in draining rainwater from roof to the ground; consists of drainage pipes with significant porosity, and drainage chambers (usually made of high density polyethylene, porous tanks with high strength and high retention capacity). An element of the drainage system can also be the much smaller drainage boxes made of grating constructed to a permissible static load (due to their small size they can be used on small plots of land). | [46,60,107,114–117] |

The current solutions in blue-green infrastructure can be divided depending on their form and location in the urban fabric into, point, linear, or surface (Figure 3). This in a systemic way could simulate and stimulate the city's natural system built with ecological corridors and ecological hubs, linear in configuration. The essence of this supporting system is not only its spatial configuration, but also its multi-scale creation process: from individual micro-realizations (small ponds, rain gardens, rainwater butts in the vicinity of a single-family house) to macro-scale solutions implemented in housing estates, districts, as well as included on master plans for entire cities.

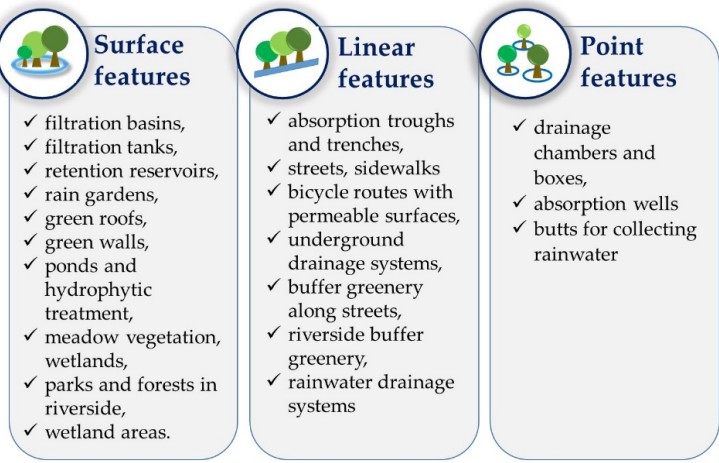

**Figure 3.** Surface-linear-point scheme for blue–green infrastructure supporting sponge cities (own elaboration).

### 3. Results: Multi-Functionality and Large-Scale Blue-Green Infrastructure Solutions Supporting Water Retention in the Urban Fabric

Urban areas, due to their specificity, are struggling with the lack of adequate green areas and this often means a water deficit. This forces an analysis of the problem and the choice of an appropriate package of solutions, especially based on blue-green infrastructure which, depending on the adopted guidelines, will stabilize the water management system to varying degrees. Introducing blue-green infrastructure into a city, in addition to its natural and environmental value supporting the water cycle, are positive economic effects connected with delivering the production of local food, which in turn reduces the dependence of cities on global supply chains. These activities should aim at the development of a closed water cycle, minimizing water losses and allowing for sustainable management of water resources. Elements of this system are based on multi-scale blue-green infrastructure, creating a patchwork of 'blue connections' that fit into the city's 'green' natural system and have a connection with urban rainwater drainage. Taking this into account, a '5-scales' system of diffusion of blue-green infrastructure, which directly and indirectly supports the design of a sponge city was established (Figure 4).

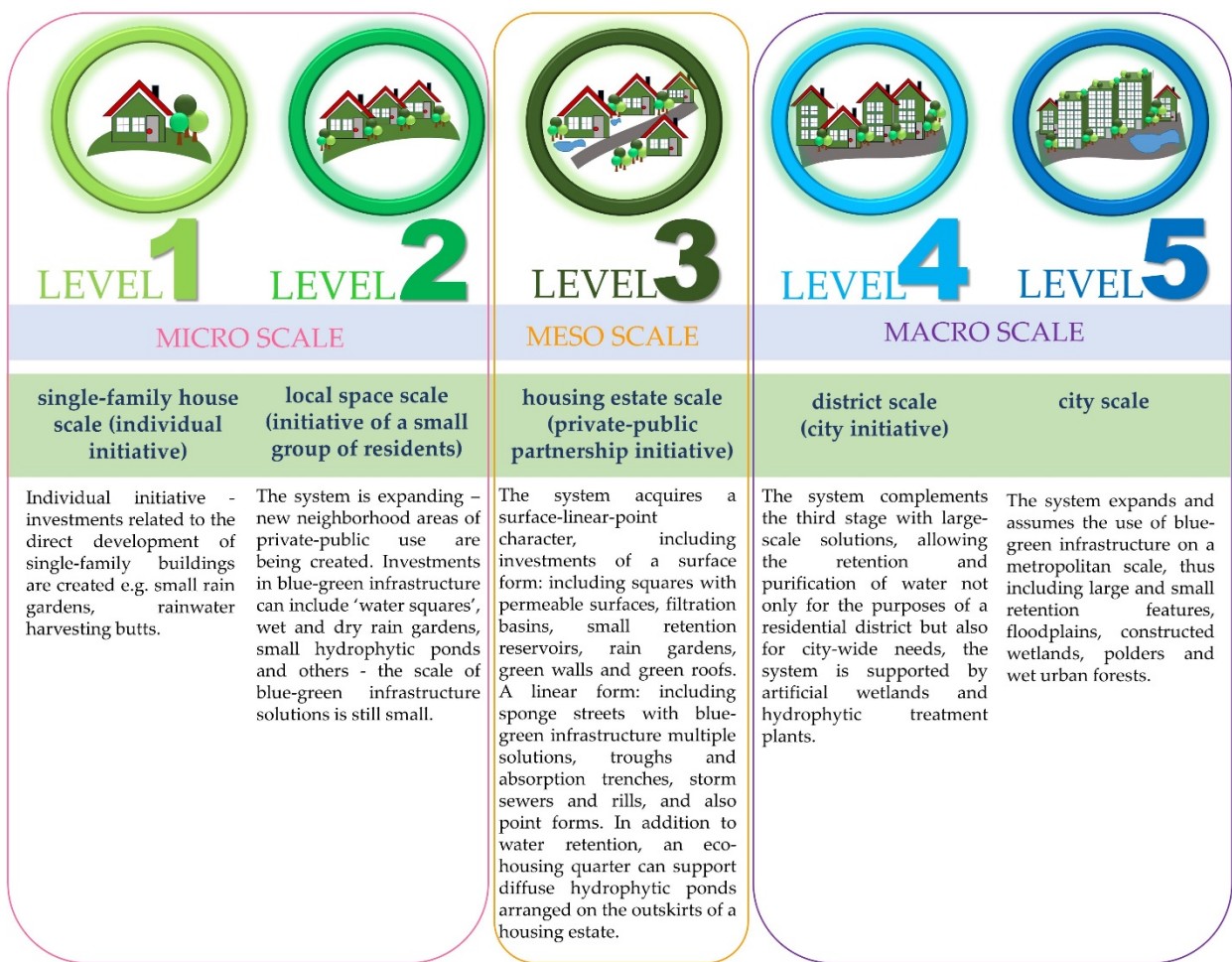

**Figure 4.** A '5-scales'system for the implementation of elements of blue-green infrastructure supporting the creation of a 'sponge city' (own elaboration).

### 3.1. Micro Scale: A Single-Family House

Single-family housing occupies significant areas which, with the introduction of innovative blue-green infrastructure, can have a large impact on the water retention of the entire city. Unfortunately, this impact is often underestimated. The creation of blue-green infrastructure should, in addition to increasing the use of biologically active areas, become

one of the obligatory elements for local spatial development plans (e.g., Cloughjordan Eco-village, Ireland; Witchcliffe Ecovillage, Australia; Beddington Zero Energy Development, UK). Excess rainwater can be used here to create new blue–green neighborhoods. These solutions may include the implementation of the following:

- butts for rainwater harvesting;
- small hydrophytic ponds and wastewater treatment plants for single-family household purposes;
- rain gardens (wet and dry);
- open vegetated trenches and small filtration basins forming ecological micro-corridors;
- filtration troughs;
- artificial and natural permeable surfaces;
- green roofs and green walls.

*3.2. Meso Scale*

When considering the role of blue-green infrastructure in the retention system in the urban fabric, it is necessary to take into account the so-called meso scale, which is formed both by surface systems, such as residential districts and downtown development centers, but also linear systems including blue-green-grey infrastructure in the form of pedestrian-bicycle routes, pedestrian routes, streets, boulevards, and avenues. When creating new eco-housing estates, they usually have a much larger area that can be developed through blue-green infrastructure in both surface and linear systems. Of fundamental importance is the use of solutions strengthening retention. Choosing the right method for using surface runoff and rainwater retention depends on the individual characteristics of an area and the investment, as well as size, possible retention volume, and the number of inhabitants.

3.2.1. Meso Scale–Sponge-Street

Project activities involving the linear development of sponge cities concern the creation of 'sponge-streets' with permeable surfaces, using diverse topography and a much wider buffer belt of vegetation. In the lowering of the terrain alongside a road, rain gardens, filtration basins, bioswales and other solutions supporting the cleaning and retaining of water from the road surface can be implemented [119]. It can be assumed that in the cities of the future such eco-corridors at various scales and sizes will be designed; these refer to the idea of a parkway, the first green eco-road system having a significant width to buffer green belts. On the eco-street scale, in relation to the blue-green infrastructure, meso-scale solutions include:

- permeable surfaces (used both for the road surface and on roadsides, pedestrian-bicycle routes, and so on);
- vegetated filtration channels, rills and open troughs (having in addition to the function of collecting water, that of its purification);
- bioswales;
- rain gardens–the first element of the rainwater drainage system (most often located by natural or artificial lowering of the terrain, purifying water coming from the streets);
- small retention reservoirs and ponds (built depending on the needs and terrain capabilities of a given area where water flowing from the streets will be retained and purified);
- underground root boxes (collecting rainwater and enabling the planting of trees in compact developments);
- buffer strips of street greenery (mainly low) reinforced with appropriately selected and adapted filtration systems.

After heavy rain, the water that flows from the streets can also be collected in underground retention tanks, and reused in a period of drought. Purified by filtration systems (absorption wells, absorption trenches), the water can be used for greenery as well as for flushing streets. Thanks to this combination, it is possible to design one of the elements of a closed or semi-closed water circulation system in the city. The functioning of such

a system is possible if the natural topographic conditions of the area have been properly used and hydro-engineering solutions have been properly selected, together along with constant monitoring and supplementing of the system with intelligent control of rainwater management using IoT technology. It should be added here that the development of new sponge-streets thanks to vegetation can have a significant impact on the improvement of environmental, social, and economic conditions, as well as undoubtedly on the aesthetic value of the street and the entire city. It should be noted, however, that in the case of the modernization of streets located inside compact downtown development, usage of most of the blue-green infrastructure solutions mentioned is often actually impossible, and usually economically unprofitable.

### 3.2.2. Meso Scale–District

Planning blue-green infrastructure on a district scale involves activities in a different perspective, integrating more space and affecting neighboring areas (e.g., Hammarby Sjöstad, Sweden; Vauban, Freiburg, Germany; Amsterdam-Noord, Netherlands; Malmö (BO01), Sweden). Technical solutions include the use of the following:

- permeable surfaces (used both on the road surface and squares, as in the courtyards);
- filtration troughs and bioswales (in addition to the function of collecting water, also possessing the function of its purification);
- hydrophytic ponds;
- wet rain gardens (allowing the collection and purification of rainwater from roofs) and dry rain gardens;
- small surface and underground retention basins from where the purified water is transported to rain gardens or used for sanitary purposes and/or for watering plants during drought;
- underground stormwater drainage system;
- underground root boxes (collecting rainwater and enabling the planting of trees in compact urban developments);
- buffer strips of street greenery;
- system of surface street gutters and rills;
- green walls;
- green roofs;
- a system of urban eco-farms, which can be located, among others, in areas of allotment gardens.

In addition, it should be emphasized that due to the surface nature of housing estates or downtown development centers, it is possible to more intensively develop surface filtration elements that strengthen retention (open retention reservoirs, filtration basins, absorption trenches or filtration troughs), as well as introducing solutions based on natural filtration systems, such as plantings in the form of riverside, moisture-loving buffer vegetation, or the establishment of flower meadows or wetlands, with properly selected species composition. The benefits of investing in blue-green infrastructure should be primarily aimed at social benefits, while respecting the natural environment and the local economy.

### Østerbro Meso-Scale Sponge District in Copenhagen (Denmark)

In Copenhagen, after the 2011 floods, a flood-warning tool called SURFF was implemented. This program takes into account forecast rainfall and compares it with a terrain model in 3D. In addition, it monitors the state of water in urban pipes. Such an analysis allows a scenario estimating how much the water level will rise after precipitation and what the effects may be. Thanks to the use of this tool in the suburban zone of Copenhagen, a unique project was created in 2016, in the Østerbro district, Klima Kvarter (climate-resistant district), which in 2016 received the Guangzhou International Award for Urban Innovation [120,121]. In this project, on an area of about 105 ha, in addition to a number of blue-green investments (e.g., Østergro rooftop farm, the Pavement Garden on Bryggervangen, Open Gardens and the Green Entrance) there were solutions that concerned rainwater management, such as 'cloudburst road', a street which, in addition to the function of a

pedestrian–vehicle–bicycle route, also performs the function of a channel carrying rainwater during heavy rain safely away from buildings. Water is carried along Tåsingegade and Kildevældsgade Streets to Østerbrogade, where it is collected and then discharged into the port. Along the roads, pavements with a permeable surface of considerable width (up to 6 m) have been created, which allows the introduction of buffer greenery intercepting and infiltrating contaminated water coming from the streets. In this district, special attention is paid to rainwater retention in situ, thanks to solutions such as filtration basins, rain gardens, and underground retention reservoirs while other water is retained in building plots [121]. The district-wide Østerbro Climate Change Adaptation Neighborhood programme aims to use green roofs, rain gardens, rain squares, and other blue-green infrastructure developments to manage rainwater by delaying run-off as well as retaining and reusing it. One example in the Østerbro district is Tåsinge Square in the eastern part of which a vast rain garden consisting of three sunken green areas of varying depth and size, planted with moisture-resistant plant communities [121,122]. Under the square there are reservoirs that retain and purify water from the roofs of nearby buildings. The purified water from the underground reservoir is led through small channels and rills on the surface of the square, from where it is discharged into a large garden located in an artificial lowering of the terrain. Such a combined and comprehensive system creates one of the key elements of the closed water cycle, and so is important for the sustainable management of water resources in future cities.

Lingang Special Area Shanghai (China)

One of the most interesting examples, and at the same time the largest of the sponge cities, is the Lingang Special Area. The project was located in the Pudong New Area of 74 km$^2$ and carried out by Shanghai Lingang New City Investment and Construction. In this urban area, design solutions include buffer green belts, riverside greenery, artificial wetlands, roof gardens, and a comprehensive use of permeable surfaces, bioswales, and raingardens [123]. Retention tanks have been installed under gardens where rainwater accumulates and is purified and water from them is used for plants and also for sanitary purposes. In Lingang, 26 neighborhoods were distinguished, in which engineering solutions assume the use of blue-green infrastructure, and a total of 36 km of roads in the city have permeable surfaces. During frequent heavy rain, water flows along the streets to artificial wetlands where it is then retained and purified, effectively creating protection against typhoons. One notable example of a blue-green infrastructure project is the 54-hectare park, the construction of which began in 2019 [123] where in addition to retaining rainwater from runoff, another function is its purification through the use of eco-engineering developments. This park is divided into four parts by two rivers flowing through the area, and in the north-eastern part of the park a 9.6-hectare wetland with several ponds and floating islands has been created. Water pumped into the park from rivers first passes through a skimmer, where surface debris is removed, and then moves on to a system of interconnected flow ponds. Water flows through several levels of filtration ponds with aquatic plants such as vallisneria and water lilies. The project assumes that during the day the system can purify a total of up to 15,000 cubic meters of water. Artificial wetlands are connected to each other by a drainage system, which also makes it possible to purify rainwater from surrounding areas. Sludge, sand, and sediments extracted from the river bed were used to change the morphology of the terrain and create new landscape elements, such as the 'Sponge city' in Lin-gang Special Area (lingang.gov.cn).

*3.3. Macro Scale–City*

From the point of view of the entire city, one of the most important functions of rainwater management is protection against the effects of extreme weather events, floods, and droughts. Engineering solutions in particular include the following:

- location of retention reservoirs, increasing the retention capacity of rivers, retaining water that has already been brought into the river as a result of direct surface runoff and

through rainwater or general sewerage systems. The vegetation along the river banks supports the removal of pollutants from rainwater while creating an attractive valley landscape of high natural value and aesthetic, educational, and recreational functions;

- filtration basins: dry or wet in the form of channels closed with a damming structure (e.g., weir) covered with vegetation;
- hydrophytic treatment and ponds, permanently and to varying degrees supplied with rainwater, primarily used for purification;
- dry reservoirs with a constant flow, with a trough in which there is water or a shallow wetland, an aesthetic element and a refuge of biodiversity while removing impurities;
- buffer vegetation planted on the banks of water bodies, often combined with biogeo-chemical barriers (e.g., in the form of gabions);
- revitalization and restoration of watercourses based on the appropriate terrain in a form as close as possible to the natural, leaving room for the river to meander [124];
- a system of synergistically interconnected and interacting multi-solutions on a meso scale (e.g., rain gardens, filtration basins, bioswales) and on a micro scale (e.g., butts for harvesting of rain water, absorption wells and many others).

### 3.3.1. Sponge City Comprehensive Planning: China

One of the first sponge cities was described by Rooijen et al. (2005), who used the term to show the potential for using rainwater flowing from the city of Hyderabad in India to irrigate agricultural crops. Shannon also used the term sponge city to describe an urban design for the city of Vinh, Vietnam, in which alternating lowland and upland belts were created to limit the effects of flooding associated with the Lam and Vinh rivers [125–127]. In 2015, the Ministry of Housing and Urban-Rural Development in China designated 16 new cities to be built according to sponge city principles [12]. The Directive on promoting Sponge City Construction states that 20% of Chinese urban areas should absorb, retain and reuse 70% of rainwater by 2020, and this percentage should increase to 80% by 2030 [7]. The demonstration areas in the pilot cities had to be larger than 15 km$^2$ and have an average annual rainfall of over 400 mm [20]. In the first stage, cities with different hydrological conditions and belonging to different geographical zones were selected, and were located in mountainous, coastal and lowland areas in different climatic zones. Each pilot city received an appropriate government subsidy for the purchase of raw materials. Engineering and construction costs were also estimated, amounting to a total of USD 85 million per year for provincial capitals (the duration of the grant is three years); USD 71 million for municipal cities directly controlled by the central government; and USD 57 million for all other cities [20]. In 2016, 14 new pilot cities were selected, and in 2017 more cities. By design, these cities were to represent an even wider range of environmental and climatic diversity, so as to be able to trace the opportunities for applying these investments in other physiographic and climatic conditions [127]. Among the pilot cities were megacities located on the coast such as Tianjin and Shanghai, but also a number of other subprovince and prefecture cities. Among the overarching goals that the so-called sponge cities were to meet were to transform water drainage and sewage control systems (by 2023) to promote sustainable drainage construction methods, to have a permeable surface area ratio at a minimum of 40% in new developments, to convert impermeable surfaces to permeable ones, to store rainwater and reduce the hydrographic peak, to rehabilitate urban water ecology, to conserve urban water resources, to increase stormwater drainage capacity, to increase public investment in drainage projects, and to infiltrate 70% of rainfall within the development areas by 2030. Eighty percent of urban areas should achieve these objectives in using the natural landscape, topography, wetlands, farmlands, woodlands, grasslands, and existing rivers and lakes to develop synergistic eco-spaces and finally to promote water conservation, recycling, and flood and water-logging resilience [128,129].

### 3.3.2. Sponge Cities in Central Europe–An Example from Poland

According to the WWF, there are about 150,000 km of rivers in Poland, of which about 20–25 percent have not been anthropogenically transformed at all or only to a small extent. This gives rise to a great hope that general hydrographic conditions will allow the potential of this natural system to be used to increase retention in urban areas. Woven into this natural system, the renatured sections of rivers and small natural city watercourses may in the future help in building the healthy conditions of sponge cities, resistant to extremes. At the same time, unfavorable statistical data indicate that in Poland there are about 1800 cubic meters of water available per capita per year, while during a drought this indicator falls to well below 1000. It should be noted here that the average amount of water per inhabitant in Europe is about 4500 cubic meters. The reasons for this are, among others, climate change and inadequate conduct of drainage activities in agriculture after World War II [130,131].

One important document that discusses blue-green infrastructure in Poland is the so-called Strategic Plan for Adaptation to Climate Change [132]. One of the priorities included in this document is to limit the developing of areas at risk of flooding and to take into account the potential impacts of other extreme phenomena caused by climate change. One of the important tasks of urban climate policy is 'Natural revitalization, including the restoration of degraded green areas and water reservoirs to their original functions, with particular emphasis on low retention in cities and replacement of compacted soil surfaces with permeable ones, taking into account in the development plans for cities and the need to increase green and water areas and ventilation corridors. In Poland, the Water Law Act pays special attention to water retention and counteracting the reduction of natural retention and the direct discharge of water from sealed areas. In accordance with the requirements of the [133], member states are obliged to introduce the so-called 'reimbursement of environmental costs' for investors regarding fees for water maintenance. These introduce new investment opportunities related to the construction of small local retention reservoirs, slowing down the outflow of water in canals and ditches (e.g., through the construction of valves or sequences of rapids). As part of the Strategic Plan for Adaptation to Climate Change, the 44 largest cities in Poland are to be transformed into sustainable cities in terms of nature, society, and the economy, fully resistant to climate change. For these cities in Poland, Urban Adaptation Plans have been developed that take into account numerous activities in the field of blue-green infrastructure [134]. One of the most important rainwater management projects in Poland is the 'Small Retention' program which includes the implementation of the local construction of small reservoirs and ponds, tree planting, restoration of small rivers and protection of wetlands. In the State Water Holding 'Polish Waters', as part of the 'Stop drought!' program, activities which are related to increasing retention in urban and rural areas and in the open landscape are being taken. In response to the needs of rainwater use, new rain gardens are being created, for example in 2019, the city of Warsaw launched the 'Warsaw for climate' program, Wrocław 'Catch the rain', and Łódź 'Rainwater harvesting'. Unfortunately, the scale is still small, although housing estates with retention are already built. Gdańsk is one of the Polish cities that actively uses five levels of retention: reservoir, terrain, street, backyard, and urban greenery.

One of the Polish cities that strives to achieve the status of a sponge city is Bydgoszcz, where intensive modernization of the existing rainwater infrastructure in the city is underway and new elements of a retention system are being built. As part of the project, a catalogue of technical solutions to encourage investors, including individual residents of the city, to save rainwater has been developed. The project was divided into two stages. The first includes the construction of new stormwater sewers with a length of approx. 14 km, the construction of 66 retention reservoirs, 22 rainwater treatment plants, and 11 outlets to receivers, as well as the construction of devices enabling the purification and management of rainwater in green areas and the reconstruction of stormwater sewers. The second stage concerns the renovation and cleaning of existing stormwater sewers with a length of approx. 90 km [135].

Another city in Poland which, due to climate threats, has joined the sponge city program is Legnica. The program related to small retention will be implemented in several stages until 2023. The first stage will include an inventory of the current green and technical infrastructure and soil conditions. The second stage effect will be the designing of ten water retention projects implemented in cooperation with residents. In the third stage, numerous ideas for collecting rainwater, water gardens, and the elimination of impermeable surfaces will be introduced. New tree plantings will be carried out and green walls, flower meadows, pocket parks, and butterfly gardens will be created [136]. Low retention is one of the most important elements of the Urban Adaptation to Climate Change Plan, which in 2019 was adopted by the City Council [137]. In Poland, numerous cross-border projects devoted to adaptation to climate change have been implemented for many years on the Polish–Saxony border. Under projects with the acronym TRANSGEA ('Cross-border cooperation on local adaptation to climate change') and WIKT (Support for climate action in a cross-border region) [138] a large database of micro-adaptive activities and a catalogue with examples of good practices in the field of blue-green infrastructure describing projects for rainwater retention and increasing biodiversity have been created, including balconies, evaporating pools, 'hotels' for insects, flower meadows, absorption troughs, tree plantings, permeable surface areas, rain gardens, oxbow lakes, ponds, and green walls [138]. In addition, for each good practice, matrices containing the purpose of the adaptation, the scale of impact and the cost of implementation have been developed. The adaptation activities carried out as part of these projects are an example of practical solutions on a local (microscale) and low-budget (micro-adaptation) scale.

## 4. Discussion

Modern urban design consists of many stages in which an important element is to increase retention and create biologically active surfaces which perform different functions depending on scale. Until recently, blue-green infrastructure had primarily an ecological function, building the city's natural system, helping increase biodiversity, and supporting the climate balance in the urban fabric. Today, to the many other benefits of creating blue-green infrastructure, the important and hitherto neglected function of food production related to the creation of urban eco-farms should be added. Food production with the support of blue-green infrastructure seems to be the next inevitable and important step towards the sustainable development of cities and improving their self-sufficiency. One solution for this purpose that has been used for a long time is the green roof [83,93,139–143]. Importantly, but unfortunately under-appreciated, areas that can support such solutions are common mainly in Europe in the so-called workers' allotment gardens [144–147], owned by the city and used by its residents. Excess water used for watering gardens and accumulated in butts can be used in surface retention, so food production, next to recreation, can become the main function of these areas (Table 2). Modern cities also have other spaces that can create elements of blue-green eco-farms. These include urban wasteland associated with wetlands, ideal for the implementation of the assumptions of a sponge city, post-industrial areas including former military land, former industrial plants with buffer green belts, or investments on inactive rail and tram lines. A system of 'horizontal' eco-farms can be supported by vertical farms, which create additional usable spaces for urban crops. It should be mentioned here that the new multi-functional urban eco-farms combine natural, social, economic, and landscape functions. On their territory, it is possible to establish shops selling local produce, restaurants, squares, greenery, recreational spaces, and much more. Partly supplied with rainwater from investment in blue-green infrastructure, they will constitute one of the new elements in the system for the entire city. It is possible that multi-functional urban farms will become elements integrating public spaces with services and food production for local communities [143].

**Table 2.** Benefits of introducing retention-maximizing solutions into design (own elaboration).

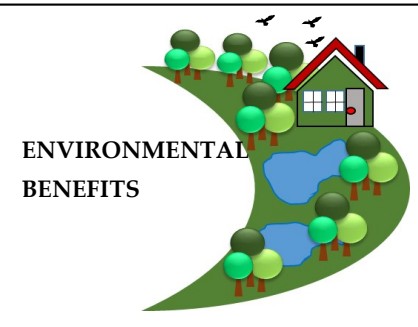

**ENVIRONMENTAL BENEFITS**

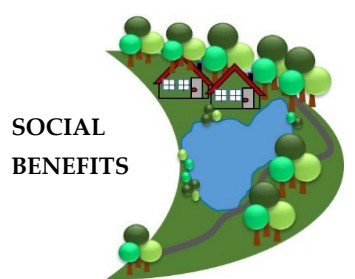

**SOCIAL BENEFITS**

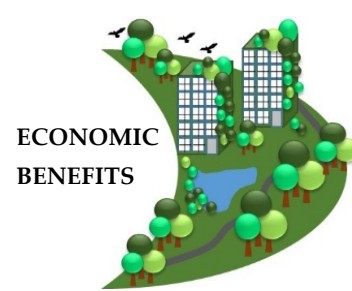

**ECONOMIC BENEFITS**

- better water quality (reduction of pollution through the use of vegetation);
- reducing the risk of flooding (increase in biologically active areas, creation of retention reservoirs, bioswales, rain gardens);
- restoration of natural rainwater circulation in nature (including buffer greenery, green roofs, green walls);
- the use of biologically active surfaces on large areas of parking lots, squares and roads (surfaces such as mineral, mineral-resin, permeable concrete, geogrids or concrete grating filled with grass or gravel);
- preservation of water resources and water retention (creation of new surface and underground reservoirs);
- reuse of rainwater (collected in tanks it can be used for watering plants during drought, for flushing streets and in residential and public buildings for flushing toilets);
- increase in biodiversity (due to the revitalization of rivers and the introduction of NBS Solutions);
- creation of new ecological corridors and habitats (in river valleys, along natural watercourses and canals, creating new ponds, bioretention gardens and retention reservoirs);
- creating new blue-green areas (excess water flowing from the roofs of buildings can be used to create solutions based on NBS);
- participation in the creation of the city's natural system (solutions in the field of blue-green infrastructure should be the basis for planning activities involving the shaping of the natural system of modern eco-cities);
- impact on the microclimate in the city (impact on reducing the negative phenomenon of the urban heat island).

- creating recreational spaces in the city (the newly created blue-green infrastructure provides a wide spectrum of recreational areas);
- the opportunity to create and support an active urban community (the development of blue-green infrastructure is conducive to shaping neighborhood spaces within housing estates);
- an opportunity for public involment in the creation of new blue-green infrastructure projects;
- creating safe urban spaces (participation of residents in the design of new multifunctional blue-green areas promotes greater social activity and reduces acts of vandalism);
- eco-education (by raising the ecological awareness of residents, facilitated by workshops, pro-ecological festivals, etc. organized at municipal and local levels);
- creation of new pedestrian and bicycle routes (using grey infrastructure alongside the blue-green e.g., the use of permeable surfaces and rain gardens purifying water from the streets in natural depressions);
- beneficial effects on the health of residents.

- lower wastewater bills (reduction in the amount of wastewater due to 'interception' of water during heavy rain);
- reduction of the operating costs of rainwater drainage systems;
- reduction of rainwater management costs;
- reduction of material losses as a result of floods and flooding;
- reduction of water deficits in the city (introduction of two-cycle, or in favourable conditions, of three-cycle use of water divided into drinking water, water for urban vegetation, and water used for urban sanitary purposes - for flushing toilets and washing streets);
- thanks to green walls and green roofs, reducing the cost of building maintenance (heating and air conditioning costs);
- an increase in land prices in areas located in the vicinity of blue-green infrastructure investments (creating new attractive environment for investors and residents of urban areas);
- the impact of blue-green infrastructure on the designation of a closed logistics and supply chain for cities through introducing new urban eco-farms (on green roofs, green walls, allotment gardens);
- thanks to investments in blue-green infrastructure (among others: green roofs, green walls), the possibility of purchasing food produced in the city.
- positive function of authorities, governments in terms of introducing retention-maximizing solutions in the cities (e.g., introducing law encouraging green investments).

Blue and green infrastructure therefore supports ecosystem services throughout the city, providing a healthy and safe environment for residents through rational use, but also with the protection of natural resources. Blue and green infrastructure also contributes significantly to reducing the cost of water treatment, as well as reducing the costs associated with local flooding and wastewater management. It should be mentioned here that due to the different needs of the city, resulting from its topography, types of existing development, intensity, and share of greenery, plans covering blue and green infrastructure should be subject to detailed evaluation in terms of their impact on the environment and biodiversity as well as the local community and economy.

## 5. Conclusions

Contemporary urban design should include multi-system solutions that can be used at multiple scales. One of the most important issues from the point of view of sustainable urban development is the construction of climate-resilient flood-protection systems that can also meet other needs of local residents. The '5-scales' system of diffusion of blue-green infrastructure determined as part of the research may therefore be the foundation for the construction of a water-oriented city, one where water infiltration and retention is at a sufficiently high level, and the biological balance, despite anthropogenic transformation, confers positive effects to create a resilient city biome. Thus, it becomes a friendly place for human life, as well as plants and animals, and prevents the formation of concrete enclaves with a reduced biologically active surface. Therefore, it seems that in the near future the overriding role of naturally sustainable cities will be not only to complement ecological corridors by creating a natural system, but above all to increase water retention obtained through the appropriate synergy of macro-, meso-, and micro-scale design of blue-green infrastructure in the urban fabric. Multi-scale solutions that increase retention in the urban fabric, in conditions of the synergistic interactions of many factors in different spaces in a positive way, can lead to the creation of sustainable urban ecosystems and positively support the city's biome. Excess water from heavy rain, which is used during droughts and for sanitary purposes, can be partially collected in underground retention tanks, as well as distributed through a drainage system including that for rainwater, drainage chambers, and drainage boxes. It should be mentioned that open vegetated reservoirs and bioswales can create an attractive space for residents, improving the microclimate and providing access to new recreational areas that can be used in various ways, e.g., to create outdoor gyms, health paths, squares, or parks. A special place in the support of blue-green infrastructure is found for allotment gardens which can constitute a basic structural element for the development of a new type of urban eco-farm.

Intensive urban development, the sealing of permeable surfaces, the conversion of biologically active areas and the regulation of rivers cause increased surface runoff, the overloading of sewage systems, canals, and rivers and, as a result, problems with flooding. In addition, in recent years, extreme climatic events have become increasingly frequent, primarily floods, but also droughts. Cities, defending themselves against these unfavorable phenomena in terms of sustainable development (ecological, social, and economic), have introduced a number of solutions that can help in the retention and infiltration of rainwater. Contemporary cities see their future as creating settlement units based on a closed cycle and natural processes. Water-oriented cities in the near future will be able not only to introduce such systems, simulating natural processes, but act as a sponge to deliver water in times of drought to places where it will be needed.

**Author Contributions:** Conceptualization, A.K. and A.Z.; methodology, A.K. and A.Z.; software, A.K., A.Z. and M.A.-P.; validation, M.A.-P., K.W. and D.v.d.H. formal analysis, A.K. and A.Z.; investigation, A.K., A.Z., M.A.-P., K.W., F.G., R.M. and D.v.d.H.; resources, A.K., A.Z., M.A.-P. and K.W., data curation: A.K. and A.Z.; writing—original draft preparation, A.K. and A.Z.; writing—review and editing, A.K., A.Z., M.A.-P., K.W., F.G., R.M. and D.v.d.H., visualisation, A.K. and A.Z.; supervision, A.K. and A.Z.; project administration, A.K. and A.Z. funding acquisition, K.W. and A.Z. All authors have read and agreed to the published version of the manuscript.

**Funding:** This research received no external funding.

**Institutional Review Board Statement:** Not applicable.

**Informed Consent Statement:** Not applicable.

**Data Availability Statement:** Data is contained within the article.

**Conflicts of Interest:** The authors declare no conflict of interest.

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
