# Peer review of "Water Oriented City—A ‘5 Scales’ System of Blue and Green Infrastructure in Sponge Cities Supporting the Retention of the Urban Fabric"

_water, doi:10.3390/w14244070_

Round 1
Reviewer 1 Report
Overall Comments
The paper reviewed an interesting and important topic. The authors had conducted an extensive review on existing studies or practice around the world in the chosen topic and delivered some comprehensive and insightful summaries. There are some areas that require polishing, restructuring, or expanding. Furthermore, referencing in the manuscript was a bit concern too. The authors needed to provide references for the statements/arguments that are not originally theirs.
Introduction
The Introduction section serves more like background or preface. Suggest including the aim and the contents/structure of the paper in the Intro section.
Suggest defining ‘blue-green infrastructure’ to help the understanding of readers who are not very familiar with the term.
Materials and Methods
Line 69, the authors intention – the authors’ intention
Line 72-82 have exactly same wording as those in Figure 1. Is it necessary to repeat it?
Line 187-251: are these the authors’ own work based on a review of literature? It wasn’t clear. Also, is it appropriate to include these discussions in section Materials and Methods?
Line 252: “3. Technologies supporting…” – incorrect heading numbering.
Figure 3: how did the authors define and group the 3 features? It needed to be explained to readers.
Results
“5-scales” system discussion and examples:
3.1 Level 1 single-family house – it would be good to include a couple of examples, i.e. such practice is seen in countries such as…
Level 2 didn’t get discussed. Suggest including some discussion on it and give a couple of examples.
Suggest revisiting the subheadings to keep consistency.
Are the solutions for each level the authors’ own work based on existing studies or practice? It wasn’t clear.
Line 485-490: repeated information. Page 4 has the exact info. Was the repeat necessary?
The China example in page 15-17 is a super long paragraph. Suggest splitting it into a few.
Line 501: Beijing is not a city located on the coast.
Line 513: please confirm the Lingang Special Area project is at the district level or city level. It is a district in Shanghai.
Discussion
The section requires an expansion. It is a bit weak now. Some of the discussions in section Materials and Methods or section Results could be relocated in section Discussion. Or, simply combine section Results and Discussion into one section.
Conclusion
It seems also a bit weak. Suggest including your key findings, recommendations or contributions to the field of research into this section.
Others
Suggest avoiding using very long paragraphs.
Suggest checking the indent of paragraphs for consistency.
Reviewer 2 Report
The reviewed manuscript raises a currently crucial issue related to water retention in the urban areas. This manuscript presents a sustainable methods of urban development with relation to application of the ‘sponge city’ concept. Moreover, the authors indicated the variable hydro-engineering solutions into the urban fabric that might allow infiltration and retention at different scales of such planning. In my opinion the discussed topic is consistent with the aim and scope of the Journal and will be of interest to the Water readers. However, at this stage, the manuscript needs minor revision.
The detailed comments are presented below:
In the abstract please indicate the most important findings of this study (the results of performed analysis). At this stage this section is too extensive and disorganized. Please redraft this section.
At the end of the introduction please highlight the novelty of the study.
In Materials, please indicate the function of authorities, governments in terms of introducing retention-maximizing solutions in the cities (line 251) e.g. legal aspects.
In the conclusions section please indicate the future perspective, actions that might be taken in spreading this assumption (it can be also presented as bullet points).
Round 2
Reviewer 1 Report
The authors well addressed my previous comments. I am happy to recommend an acceptance for publication.